# Maternal Anemia during the First Trimester and Its Association with Psychological Health

**DOI:** 10.3390/nu14173505

**Published:** 2022-08-25

**Authors:** Dong-Wook Kwak, Seokyung Kim, Su-Young Lee, Min-Hyoung Kim, Hee-Jin Park, You-Jung Han, Dong-Hyun Cha, Moon-Young Kim, Jin-Hoon Chung, Bumhee Park, Hyun-Mee Ryu

**Affiliations:** 1Department of Obstetrics and Gynecology, Ajou University School of Medicine, Suwon 16499, Korea; 2Department of Psychiatry, Myongji Hospital, Hanyang University College of Medicine, Goyang 10475, Korea; 3Department of Obstetrics and Gynecology, MizMedi Hospital, Seoul 07639, Korea; 4Department of Obstetrics and Gynecology, CHA Gangnam Medical Center, CHA University, Seoul 06135, Korea; 5Department of Obstetrics and Gynecology, Asan Medical Center, University of Ulsan College of Medicine, Seoul 05505, Korea; 6Department of Biomedical Informatics, Ajou University School of Medicine, Suwon 16499, Korea; 7Department of Obstetrics and Gynecology, CHA Bundang Medical Center, CHA University, Seongnam 13496, Korea

**Keywords:** anemia, anxiety, depression, first trimester of pregnancy

## Abstract

Anemia during pregnancy is known to be associated with an increased risk of antenatal and/or postnatal depression, as well as adverse pregnancy outcomes. However, there are few studies evaluating psychological health throughout the antepartum and postpartum periods in women with anemia in early pregnancy. This study analyzed data collected by the Korean Pregnancy Outcome Study, a multicenter prospective cohort study conducted in South Korea, to determine the impact of anemia during the first trimester on birth outcomes and maternal mental health during pregnancy and postpartum. Hemoglobin levels were measured during the first trimester, and psychological health was evaluated at 12, 24, and 36 gestational weeks and 4–6 weeks postpartum. Anxiety and depression were defined using the Hospital Anxiety and Depression Scale and the Edinburgh Postnatal Depression Scale, respectively. Among 4067 Korean participants, 119 (2.9%) were diagnosed with anemia during the first trimester. Incidences of anxiety and depression did not differ over the pregnancy period between those with and without anemia during the first trimester. However, postpartum anxiety and depression were significantly more common in participants with anemia than in those without (*p* < 0.05, both). Hence, obstetricians should pay attention to postpartum mental health in women with anemia during the first trimester.

## 1. Introduction

Anemia is characterized by a lower-than-normal number of red blood cells or hemoglobin (Hb) concentration within them [1]. Anemia is one of most frequent complications related to pregnancy. Several studies have been published regarding anemia during pregnancy, and it is known to be associated with an increased risk of adverse pregnancy outcomes, such as premature delivery, low birth weight, and small-for-gestational-age (SGA) neonates [2,3,4,5]. However, the prevalence of anemia is very different by maternal ethnicity and socioeconomic status. Therefore, research investigating the relationship between anemia and pregnancy outcomes across diverse ethnic groups is still needed.

In addition to birth outcomes, anemia during pregnancy can affect maternal mental health during the pregnancy and/or postpartum period [6,7,8,9,10,11,12,13]. Although several studies have reported no significant relationship between anemia and depression in pregnant individuals [14,15,16,17], a recent meta-analysis concluded that anemia during pregnancy is significantly associated with an increased risk of antepartum and postpartum depression [13]. The presence of antepartum depression has not been found to be correlated with postpartum depression, and depression can occur in any trimester of pregnancy, as well as during the postpartum period [18]. Therefore, the impact of anemia during pregnancy on psychological health might be different, not only before and after delivery, but also between each trimester. However, there is no study to analyze the presence of anxiety or depressive symptoms throughout the antepartum and postpartum periods in women with and without anemia in early pregnancy.

The Korean Pregnancy Outcome Study (KPOS) is a large-scale prospective cohort study conducted in South Korea. The study aimed to investigate the prevalence of pregnancy complications among South Korean women and to determine their risk factors by collecting information from both questionnaires and clinical data. The purpose of our study is to investigate the prevalence of and risk factors for anemia during the first trimester in the KPOS cohort. Additionally, using the KPOS data, we aim to determine the impact of anemia during the first trimester on adverse pregnancy outcomes, including psychological health during pregnancy and postpartum.

## 2. Materials and Methods

### 2.1. Study Design and Participants

This study is an analysis of data that were collected from the KPOS, which was a prospective cohort study performed at the Cheil General Hospital & Women’s Healthcare Center and CHA Gangnam Medical Center between 2013 and 2017 to investigate the prevalence of diverse pregnancy-related complications and their risk factors among pregnant Korean individuals (Trial registration: kct0000776. https://cris.nih.go.kr/cris/search/detailSearch.do/21946 (accessed on 25 June 2013). The study was approved by the institutional review boards (IRBs) of Cheil General Hospital (IRB number: CGH-IRB-2013-10) and CHA University (IRB number: 2013-14-KNC13-018), respectively. All participants provided written informed consent, and it was emphasized to all participants that they were free to withdraw from any part of the study at any point in time.

All participants were invited to attend five appointments: Visit 1 at approximately 12 gestational weeks (GWs), Visit 2 at approximately 24 GWs, Visit 3 at approximately 36 GWs, Visit 4 at delivery, and Visit 5 at 4–6 weeks postpartum. At each visit, all participants were asked to complete a self-administered questionnaire under the supervision of trained interviewers, and clinical assessments, such as the measurement of maternal body weight and blood pressure, were performed.

During the study period, all pregnant women who visited Cheil General Hospital & Women’s Healthcare Center and CHA Gangnam Medical Center for antenatal care during the first trimester were asked to participate in the KPOS. The inclusion criteria were ethnically Korean women with singleton or twin pregnancies. During the study period, 4537 participants were recruited. Of these, 342 were excluded (10 who were foreigners, 114 who had missing key variables, and 218 who withdrew) in the KPOS. More details about the study design and participants of KPOS, including sample size calculation, were described in our previous study [19].

### 2.2. Variables

Demographic information, including maternal age, educational level, house income and marital status, was assessed at Visit 1, and change in marital status was evaluated at each visit. Body mass index was calculated based on the maternal height and body weight at Visit 1 and categorized according to the Asian-Pacific cut-off points for weight defined by the World Health Organization (WHO) [20]. The obstetric information, such as parity, number of fetuses, whether the patients had planned pregnancy, and the conception method, was obtained from prenatal records. Cigarette smoking (current smoking, duration and amount), alcohol intake (current drinking, duration, and amount), and supplement intake were evaluated at each visit. The participants who took the iron pills and/or multiple micronutrient (MM) supplements were defined as the iron supplementation group. Hyperemesis (nausea and vomiting in pregnancy) and threatened abortion, which was defined as pregnancy-related bloody vaginal discharge or frank bleeding during the first half of pregnancy without cervical dilatation, were assessed at Visit 1.

Hb levels were evaluated at Visits 1 and 3, and anemia was defined as a Hb level of <11.0 g/dL, according to the WHO classification [21]. Psychological health was evaluated at Visits 1, 2, 3, and 5. Anxiety was determined using the Hospital Anxiety and Depression Scale (HADS). The HADS for the anxiety subscale comprises seven questions to measure anxiety symptoms, and each question is scored on a response scale with four alternatives ranging between 0 and 3 (range 0–21). A total score of ≥8 is considered the cut-off point for an anxiety diagnosis [22]. Depression was assessed using the Korean version of the Edinburgh Postnatal Depression Scale (K-EPDS). The K-EPDS comprises 10 questionnaires (range 0–30) designed to screen for possible depression, and a total score of ≥10 is considered the cut-off point according to a previous South Korean validation study [23].

Pregnancy outcome variables included gestational age at birth, birth weight, preterm delivery, abnormal birth weight such as low birth weight or SGA <10th centile [24], cesarean section rate, low Apgar score, and hypertensive disorders of pregnancy.

### 2.3. Statistical Methods

Values are presented as frequencies (percentages) or medians (interquartile ranges), as appropriate. The chi-square test and Wilcoxon rank sum test were used for categorical and continuous variables, respectively, to compare maternal characteristics and pregnancy outcomes, including psychological health between women with and without anemia. To identify risk factors for anemia during the first trimester and to determine the risk of anemia in early pregnancy giving rise to adverse outcomes, the odds ratios (ORs) and 95% confidence intervals (CIs) were assessed using multivariable logistic regression analysis. In all tests, a threshold of *p* < 0.05 was considered statistically significant. Statistical analyses were performed using IBM SPSS ver. 25.0 (IBM Corp., Armonk, NY, USA) and R 3.4.1 (Vienna, Austria; https://www.R-project.org/ (accessed on 23 April 2022)).

## 3. Results

A flow diagram of the study is shown in Figure 1. Of the 4195 participants who met the inclusion criteria of the KPOS, 128 were excluded in this study because complete blood count was not performed on them at Visit 1. The prevalence of anemia during the first trimester was 2.9% (119 patients). Among these patients, 91 (76.5%) had mild anemia, with an Hb level of ≥10 g/dL [25]. In contrast, the prevalence of anemia during the third trimester was 8.7% (297/3404), and the proportion of mild anemia was 79.1% (235/297). The distribution of Hb concentration during the first trimester in participants of the KPOS cohort is demonstrated in Figure 2.

The general characteristics of the participants according to the presence or absence of anemia during the first trimester are listed in Table 1. Among these characteristics, not having a marital partner (adjusted OR (aOR): 2.84, 95% CI: 1.39–5.80; *p* = 0.004) and twin pregnancies (aOR: 3.38, 95% CI: 1.26–9.04; *p* = 0.015) were identified as independent risk factors for having anemia during the first trimester in this study (Table 2). In addition, the incidence of anemia tended to increase with increasing maternal age, and a trend towards a higher risk of anemia was observed in women with maternal age > 40 years (aOR: 1.95, 95% CI: 0.98–3.86; *p* = 0.06).

After excluding participants with twin pregnancies and medical problems, such as hypertension or pregestational diabetes, we compared the intakes of iron or MM supplementation and pregnancy outcomes and intakes of iron or MM during pregnancy in participants with and without anemia during the first trimester (Table 3). In this analysis, the median birth weight was significantly lower, and the incidence of low birth weight and SGA neonates was significantly higher in patients with anemia than in those without (*p* = 0.02 and 0.001, respectively). Table 4 shows the changes in anxiety and depressive symptoms from the first trimester until the postpartum period, according to the presence or absence of anemia during the first trimester. The incidences of anxiety and depression did not differ significantly between the patients with and without anemia during the entire pregnancy period. Moreover, the incidences of anxiety and depression during the third trimester were lower in patients with anemia during the first trimester than in those without. However, postpartum anxiety and depression were significantly more common in patients with anemia during the first trimester (*p* < 0.05, both).

After adjusting for confounding factors such as maternal characteristics and adverse pregnancy outcomes, we calculated the risk of anemia during the first trimester giving rise to adverse outcomes (Table 5). Anemia during the first trimester was significantly associated with an increased risk of low birth weight (aOR: 2.29, 95% CI: 1.06–4.94; *p* = 0.03) and SGA neonates (aOR: 2.46, 95% CI: 1.41–4.29; *p* = 0.001), and tended to be associated with postpartum anxiety and depression (aOR: 1.81, 95% CI: 0.95–3.48; *p* = 0.07 and aOR: 1.61, 95% CI: 0.93–2.80; *p* = 0.09, respectively).

Additionally, we investigated the effect of iron supplementation on the prevention of adverse pregnancy outcomes in participants with and without anemia during the first trimester (Table 6). Among women with normal Hb levels, the incidence of low birth weight was significantly lower in participants who received iron pills or MM supplements during the first trimester than in those who did not. Similarly, among women with anemia, iron or MM supplementation during the first trimester seemed to have a positive effect on low birth weight and SGA neonates. Whereas, in women with normal Hb levels, the incidence of postpartum anxiety and depression was not significantly different between the participants with and without iron or MM supplementation. Furthermore, the positive effect of iron or MM supplementation on postpartum anxiety and depression was not observed in women with anemia during the first trimester.

## 4. Discussion

We examined the prevalence and risk factors for anemia during the first trimester in the KPOS cohort, and estimated the risks of anemia during the first trimester for adverse pregnancy outcomes, including psychological health during pregnancy and postpartum.

According to the report of the WHO (2019), the global prevalence of anemia was 36.5% among pregnant women [26]. In their report, the prevalence of anemia among pregnant South Korean women was 14.7%. In contrast, the prevalence of anemia during the first and third trimester in the KPOS cohort was 2.9% and 8.7%, respectively. The KPOS was performed at two famous maternity hospitals in Seoul, the main city of South Korea. Accordingly, the KPOS participants tended to have a higher socioeconomic status than the overall South Korean national population [19]. This may have contributed to the low prevalence of anemia in our study. However, a low prevalence of anemia during the first trimester is not a characteristic unique to the KPOS cohort. In an analysis of an international multicenter prospective cohort consisting of low-risk nulliparous pregnant individuals, the prevalence of anemia in early pregnancy was reported to be 2.2% [27]. In addition, in a cohort study in China, the overall prevalence of anemia during pregnancy was 23.5%, but its prevalence specifically in the first trimester was 2.7% [28]. During pregnancy, plasma volume expansion begins as early as 6 weeks of gestation, peaks around 32 weeks of gestation, and plateaus until delivery [29]. Accordingly, Hb levels in pregnant individuals tend to decrease with advancing gestational age. For this reason, the prevalence of anemia in the first trimester is likely to be lower than that in the second or third trimester [30,31,32].

Low income, low education status, teenage pregnancy, advanced maternal age, multiparity, pre-pregnancy underweight, having no marital partner, multiple gestation and experiencing severe nausea or vomiting during pregnancy have previously been suggested as factors potentially associated with anemia in pregnancy [27,33,34,35,36,37,38,39,40]. In our study, having no marital partner and twin pregnancy were found to be independent risk factors for anemia during the first trimester. A prospective longitudinal study demonstrated that the prevalence of anemia in multiple gestation was significantly increased from the first trimester, and early anemia was predictive of anemia during later gestation [39]. Moreover, multiple pregnancy is a major risk factor for postpartum hemorrhage [41,42]. Therefore, it is important for women with multiple pregnancies to maintain normal Hb levels until delivery. One randomized controlled study reported that doubling the dose of iron in twin pregnancies complicated by iron deficiency anemia increased Hb and ferritin levels without worsening gastrointestinal side effects [43].

The WHO recommends the definition of severe, moderate, and mild anemia for pregnant women as Hb concentrations of <7.0 g/dL, 7.0 to 9.9 g/dL, and 10.0 to 10.9 g/dL, respectively [25]. In previous studies, moderate and severe anemia has been consistently shown to have an adverse effect on pregnancy outcome, whereas the impact of mild anemia was controversial [44,45,46,47,48,49]. Furthermore, several studies showed that mild anemia during pregnancy was associated with decreased risks of adverse pregnancy outcomes [47,48,49]. This might be related to the failure of plasma volume expansion during pregnancy. A few studies indicated that a smaller reduction in Hb levels from early pregnancy to late pregnancy may suggest a failure to expand plasma volume during pregnancy [50,51,52]. The failure of maternal plasma volume expansion has been implicated in adverse pregnancy outcomes such as pre-eclampsia, fetal growth restriction and preterm birth [53]. Therefore, the clinical significance of mild anemia in late pregnancy might be different compared to that in early pregnancy. In the KPOS cohort, the majority of anemic participants during the first trimester had mild anemia, but low birth weight, or SGA was significantly more common in participants with anemia. Obstetricians should consider our findings when counseling patients diagnosed with mild anemia in early pregnancy.

Many studies have investigated psychological health in women with anemia during pregnancy [6,7,8,9,10,11,12,13,14,15,16,17,18]. However, there are few studies evaluating the impact of anemia in early pregnancy on antenatal or postnatal mental health. Armony et al. conducted an observational study to evaluate the relationship between pre- and postnatal maternal iron status and depressive symptoms in Chinese women [15]. In their study, the authors could not find any significant association between anemia in early pregnancy (between 13 and 20 weeks) and postpartum depression. However, the study was limited by its small sample size. In our study, postpartum anxiety and depression were found to be significantly more common in patients with anemia during the first trimester than in those without. Even though the significance of this difference was not observed when the results were adjusted by maternal characteristics and pregnancy outcomes, the small number of participants who were diagnosed with anemia in the KPOS cohort should be taken into account. Therefore, we cannot exclude the possibility that maternal anemia during the first trimester might be an independent risk factor for postpartum anxiety and depression. Based on our findings, we suggest that the mental health of the patients with anemia during the first trimester may be more vulnerable to physiological changes after delivery including decreased Hb levels due to intrapartum hemorrhage.

In contrast, we found that the incidences of anxiety and depression during the entire pregnancy period did not differ according to the presence or absence of anemia during the first trimester. Several studies have demonstrated that anemia during pregnancy is associated with an increased risk of antenatal depression [6,7,8,9]. However, the studies evaluated the mental health status of pregnant individuals who were diagnosed with anemia during late pregnancy. As discussed above, the clinical significance of anemia can be different between early and late pregnancy. Moreover, the large proportion of mild anemia in the KPOS cohort could be the reason for the negative result in our study. Further studies are required to evaluate the impacts of anemia on mental health during pregnancy and the postpartum period, both in early pregnancy and late pregnancy, in cohorts that have a much higher prevalence of moderate or severe anemia.

It is undeniable that iron supplementation is essential for pregnant women with anemia. However, routine iron supplementation in non-anemic women is still contentious [21,54,55]. The Centers for Disease Control and Prevention (CDC) recommends that all pregnant women begin low-dose iron supplementation at the first prenatal visit [21]. In contrast, the United Kingdom guidelines concluded that there is insufficient evidence on the benefits and potential hazards of routine iron supplementation for all pregnant women [54]. The Nutrition Societies of Germany, Austria and Switzerland recommend a daily iron intake of 30 mg for pregnant women from the second trimester [55]. Unfortunately, there is no guideline on iron supplementation for pregnant women in South Korea. Nevertheless, the majority of South Korean women take iron supplements from the second trimester of pregnancy.

In the KPOS cohort, more than 90% of the participants took iron pills or MM during the second and third trimesters, but only 39.4% did during the first trimester (Table 1 and Table 3). For this reason, we compared the incidence of adverse pregnancy outcomes between the participants who received and those who did not receive iron supplementation during the first trimester to evaluate the effect of iron supplementation on pregnancy outcomes and maternal mental health. KPOS researchers did not evaluate the contents of MM, but most MM supplements for pregnant women in South Korea contain an appropriate amount of iron. In this analysis, among women with normal Hb levels, the incidence of low birth weight was significantly lower in patients with iron or MM supplementation during the first trimester. Although we cannot certainly conclude that the effect was only from the iron intake during the first trimester, our results support the CDC recommendation that all pregnant women begin iron supplementation from the first prenatal visit. Regarding the effect of iron supplementation for the prevention of postpartum anxiety and depression, our analysis showed that postpartum anxiety and depression were not associated with iron or MM supplementation during the first trimester in both women with and without anemia during the first trimester. However, a well-designed study is needed to assess the effect of iron supplementation on psychological health.

To our knowledge, this is the first study to evaluate the risks of anemia during the first trimester for mental health both during the pregnancy and postpartum periods. The main strength of this study is that the data were collected with a prospective design and large sample size. Additionally, we analyzed data on mental health scores collected at multiple time points, which enabled us to demonstrate the anxiety and depressive symptoms during the entire pregnancy among participants with and without anemia. In contrast, a limitation of this study is that the proportion of participants who were lost to follow-up or did not respond to the questionnaires regarding mental health was relatively high in the KPOS cohort. Unfortunately, Middle East respiratory syndrome broke out in South Korea during the study period, which may be one of the reasons for this.

## 5. Conclusions

Despite the low prevalence and mild severity of anemia during the first trimester in the KPOS cohort, maternal anemia was significantly associated with an increased risk of low birth weight and SGA neonate. In addition, anemia during the first trimester did not affect mental health during the entire pregnancy period; however, a trend towards a higher risk of postpartum anxiety and depression was observed in patients with anemia during the first trimester. Therefore, obstetricians should pay particular attention to pregnant patients with anemia during the first trimester, not only for fetal growth in prenatal care, but also for maternal mental health after delivery.

## Figures and Tables

**Figure 1 nutrients-14-03505-f001:**
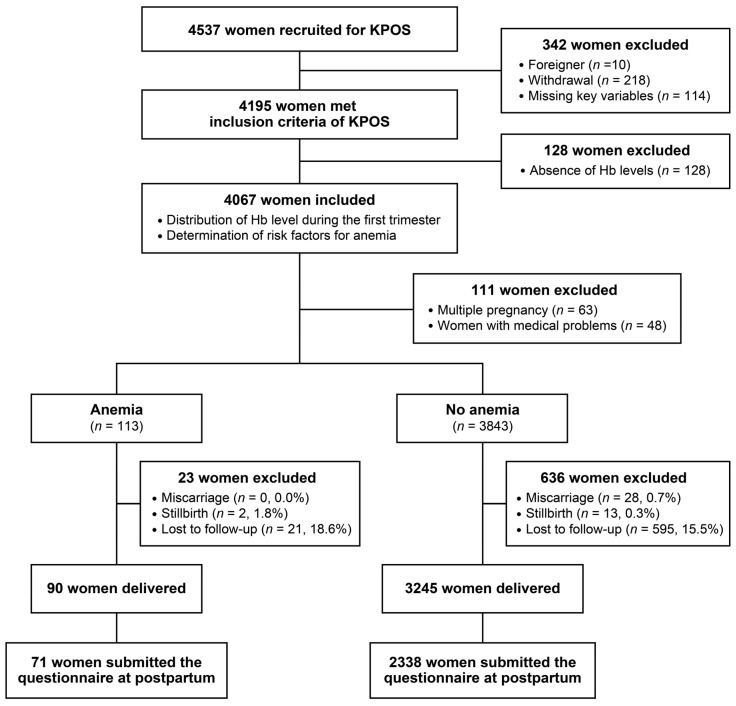
Flow diagram of this study. Hb—hemoglobin; KPOS—Korean Pregnancy Outcome Study; Miscarriage—fetal death before the 20th week of gestation; Stillbirth—fetal death after the 20th week of gestation.

**Figure 2 nutrients-14-03505-f002:**
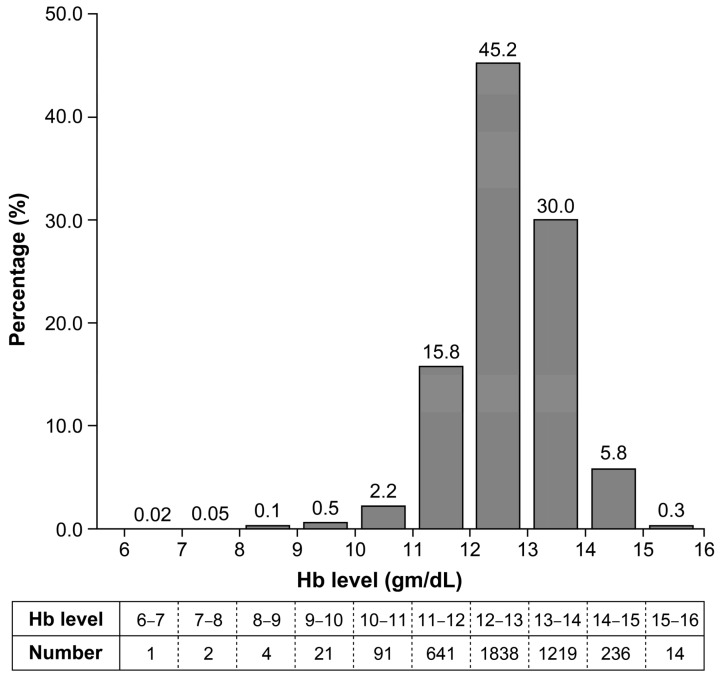
The distribution of hemoglobin level during the first trimester in Korean Pregnancy Outcome Study cohort. Hb—hemoglobin.

**Table 1 nutrients-14-03505-t001:** Characteristics of the study participants according to the presence or absence of anemia during the first trimester.

	Maternal Anemia during the First Trimester		*p*-Value
	No (*n* = 3948)	Yes (*n* = 119)	
Maternal age (years)			
<35	2535 (64.2)	70 (58.8)	0.18
35–39	1194 (30.2)	38 (31.9)	
>40	219 (5.5)	11 (9.2)	
BMI (kg/m^2^)			
<18.5	399 (10.1)	14 (11.8)	0.38
18.5–22.9	2510 (63.6)	79 (66.4)	
23.0–24.9	530 (13.4)	17 (14.3)	
>25	509 (12.9)	9 (7.6)	
Having a marital partner			
No	159 (4.0)	12 (10.1)	0.002
Yes	3789 (96.0)	107 (89.9)	
Parity			
0	2449 (62.0)	70 (58.8)	0.74
1	1307 (33.1)	42 (35.3)	
≥2	192 (4.9)	7 (5.9)	
Educational status			
≤High school	340 (8.6)	12 (10.1)	0.54
College	2935 (74.3)	91 (76.5)	
Graduate school	673 (17.0)	16 (13.4)	
Household income (won/month)			
≤3 million	510 (12.9)	20 (16.8)	0.21
3–5 million	1550 (39.3)	51 (42.9)	
>5 million	1888 (47.8)	48 (40.3)	
Planned pregnancy			
No	1519 (38.5)	53 (44.5)	0.18
Yes	2428 (61.5)	66 (55.5)	
Type of pregnancy			
Natural	3754 (95.1)	109 (91.6)	0.08
Artificial	193 (4.9)	10 (8.4)	
Twin pregnancy			
No	3890 (98.6)	113 (95.0)	0.003
Yes	57 (1.4)	6 (5.0)	
Cigarette smoking			
Never smoked	3516 (89.1)	105 (88.2)	0.80 *
Quit before pregnancy	318 (8.1)	8 (6.7)	
Quit after pregnancy	108 (2.7)	12 (5.0)	
Current smoker	5 (0.1)	0 (0.0)	
Alcohol drinking			
Never drank	768 (19.5)	29 (24.4)	0.31 *
Quit before pregnancy	1900 (48.1)	58 (48.7)	
Quit after pregnancy	1274 (32.3)	32 (26.9)	
Current drinker	5 (0.1)	0 (0)	
NVP			
No	977 (24.8)	22 (18.5)	0.12
Yes	2970 (75.2)	97 (81.5)	
Threatened abortion			
No	3243 (82.2)	101 (84.9)	0.45
Yes	704 (17.8)	18 (15.1)	
Folic acid intake			
No intake	449 (11.4)	13 (10.9)	0.47
From recognition of pregnancy	1651 (41.8)	50 (42.0)	
Before pregnancy	1643 (41.6)	46 (38.7)	
- No response	205 (5.2)	10 (8.4)	
Iron or MM intake			
No intake	2268 (57.5)	63 (56.7)	0.16
From recognition of pregnancy	961 (24.3)	35 (29.4)	
Before pregnancy	514 (13.0)	11 (9.2)	
- No response	205 (5.2)	10 (4.7)	

BMI—body mass index; FA—folic acid; MM—multiple micronutrients; NVP—nausea and vomiting during pregnancy. Data are presented as n (%). The *p*-values were calculated using chi-squared test. * *p*-value calculated using Fisher’s exact test.

**Table 2 nutrients-14-03505-t002:** Factors associated with anemia during the first trimester from univariate and multivariate analyses.

	Anemia (%)	Unadjusted OR (95% CI)	*p*	Adjusted OR * (95% CI)	*p*
Maternal age (years)					
<35	2.7	Ref.	-	Ref.	-
35–40	3.1	1.13 (0.74–1.73)	0.56	1.12 (0.72–1.74)	0.60
>40	4.8	1.94 (1.01–3.73)	0.05	1.95 (0.98–3.86)	0.06
BMI (kg/m^2^)					
<18.5	3.4	1.26 (0.70–2.25)	0.71	1.24 (0.69–2.25)	0.46
18.5–22.9	3.1	Ref.	-	Ref.	-
23.0–24.9	3.1	1.15 (0.67–1.97)	0.94	1.11 (0.64–1.92)	0.69
>25	1.7	0.64 (0.32–1.30)	0.22	0.57 (0.28–1.17)	0.13
Having a marital partner					
No	7.0	2.69 (1.41–5.12)	0.002	2.84 (1.39–5.80)	0.004
Yes	2.7	Ref.	-	Ref.	-
Parity					
0	2.8	Ref.	-	Ref.	-
1	3.1	1.11 (0.74–1.67)	0.62	1.28 (0.82–2.01)	0.28
≥2	3.5	1.26 (0.54–2.96)	0.59	1.23 (0.50–3.07)	0.65
Type of pregnancy					
Natural	2.8	Ref.		Ref.	-
Artificial	4.9	1.95 (1.00–3.80)	0.049	1.55 (0.71–3.41)	0.26
Twin pregnancy					
No	2.8	Ref.		Ref.	-
Yes	9.5	3.77 (1.59–8.94)	0.003	3.38 (1.26–9.04)	0.015
NVP					
No	2.2	Ref.		Ref.	-
Yes	3.2	1.47 (0.89–2.43)	0.12	1.51 (0.91–2.51)	0.11

BMI—body mass index; CI—confidence interval; OR—odds ratio; NVP—nausea and vomiting in pregnancy. * Adjusted for age, body mass index, having a marital partner, parity, education, household income, planned pregnancy, type of pregnancy, smoking, alcohol, nausea and vomiting in pregnancy, and threatened abortion.

**Table 3 nutrients-14-03505-t003:** Comparison of pregnancy outcomes between women with and without anemia during the first trimester.

	Maternal Anemia during the First Trimester		*p*-Value
	Yes (*n* = 90)	No (*n* = 3245)	
Hb measurements during the first trimester			
GA at sampling (weeks)	12.2 (12.0, 12.7)	12.2 (12.0, 12.7)	0.49
Hb level in the first trimester (g/dL)	10.5 (9.9, 10.7)	12.7 (12.2, 13.2)	<0.001
Iron or multiple micronutrients supplementation			
Intake during the first trimester	34/82 (41.4)	1200/3081 (38.9)	0.64
- No response	8/90 (8.9)	164/3245 (5.1)	-
Intake during the second trimester	86/88 (96.6)	3017/3103 (97.2)	0.91
- No response	2/90 (2.2)	142/3245 (4.4)	-
Intake during the third trimester	84/84 (100.0)	2892/2942 (97.1)	0.40
- No response or not available	6/90 (6.6)	303/3245 (9.3)	-
Pregnancy outcomes			
GA at birth (weeks)	39.3 (38.5, 40.1)	39.4 (38.5, 40.2)	0.49
Birth weight (kg)	3.13 (2.91, 3.40)	3.26 (3.01, 3.52)	0.001
Preterm delivery	4 (4.4)	157/3245 (4.8)	0.86
Low birth weight	8 (8.9)	119/3245 (3.7)	0.02
SGA less than 10th centile	17 (18.9)	265/3245 (8.2)	<0.001
Cesarean section	42 (46.7)	1274/3245 (39.3)	0.16
Low Apgar score at 5 min < 7	3 (3.3)	61/3245 (1.9)	0.25
Hypertensive disorders of pregnancy	3 (3.3)	37/3245 (1.1)	0.09

GA—gestational age; Hb—hemoglobin; NICU—neonatal intensive care unit; SGA—small for gestational age. Data are presented as n (%) or median (interquartile range). The *p*-value was calculated using the chi-squared test and Wilcoxon rank sum test, as appropriate.

**Table 4 nutrients-14-03505-t004:** Comparison of mental health during pregnancy and postpartum between participants with and without anemia during the first trimester.

	Maternal Anemia during the First Trimester		*p*-Value
	Yes	No	
Visit 1 (first trimester)	**(*n* = 109)**	**(*n* = 3714)**	
Depression score from K-EPDS	6.00 (3.00, 9.00)	6.00 (3.00, 9.00)	0.96
K-EPDS ≥ 10	22 (20.2)	713 (19.2)	0.79
Anxiety score from HADS	4.00 (2.00, 6.00)	4.00 (2.00, 6.00)	0.64
HADS Anxiety ≥ 8	17 (15.6)	398 (10.7)	0.11
Visit 2 (second trimester)	**(*n* = 100)**	**(*n* = 3274)**	
Depression score from K-EPDS	4.50 (3.00, 7.00)	5.00 (3.00, 7.00)	0.99
K-EPDS ≥ 10	14 (14.0)	441 (13.5)	0.87
Anxiety score from HADS	3.00 (2.00, 6.00)	3.00 (2.00, 5.00)	0.71
HADS Anxiety ≥ 8	8 (8.0)	247 (7.5)	0.86
Visit 3 (third trimester)	**(*n* = 82)**	**(*n* = 2840)**	
Depression score from K-EPDS	5.00 (3.00, 7.00)	5.00 (3.00, 8.00)	0.69
K-EPDS ≥ 10	9 (11.0)	395 (13.9)	0.45
Anxiety score from HADS	4.00 (2.00, 5.00)	4.00 (2.00, 6.00)	0.56
HADS Anxiety ≥ 8	5 (6.1)	254 (8.9)	0.37
Visit 5 (postpartum)	**(*n* = 71)**	**(*n* = 2338)**	
Depression score from K-EPDS	5.00 (2.00, 10.00)	5.00 (2.00, 8.00)	0.42
K-EPDS ≥ 10	18 (25.4)	380 (16.3)	0.042
Anxiety score from HADS	3.00 (1.00, 6.00)	3.00 (1.00, 5.00)	0.16
HADS Anxiety ≥ 8	12 (16.9)	217 (9.3)	0.031

HADS—Hospital Anxiety and Depression Scale; K-EPDS—Korean version of the Edinburgh Postnatal Depression Scale. Data are given as *n* (%) or medians (interquartile ranges). *p*-value was calculated using the chi-squared test and Wilcoxon rank sum test, as appropriate.

**Table 5 nutrients-14-03505-t005:** Risks of adverse outcomes due to anemia during the first trimester.

	Unadjusted OR (95% CI)	*p*-Value	Adjusted OR (95% CI)	*p*-Value
Low birth weight	2.56 (1.12–5.42)	0.013	2.29 (1.06–4.94)	0.034 ^†^
Small for gestational age	2.62 (1.52–4.51)	<0.001	2.46 (1.41–4.29)	0.001 ^†^
Hypertensive disorders of pregnancy	2.99 (0.90–3.02)	0.07	2.69 (0.72–10.00)	0.14 ^†^
K-EPDS ≥ 10 (postpartum)	1.75 (1.01–3.02)	0.044	1.61 (0.93–2.80)	0.092 ^‡^
HADS-A ≥ 8 (postpartum)	1.99 (1.05–3.76)	0.034	1.81 (0.95–3.48)	0.072 ^‡^

CI—confidence interval; HADS-A—anxiety score of Hospital Anxiety and Depression Scale; K-EPDS—Korean version of the Edinburgh Postnatal Depression Scale; OR—odds ratio. ^†^ Adjusted for maternal age, parity, body mass index, having a marital partner, parity, household income, and gestational weight gain. ^‡^ Adjusted for maternal age, parity, body mass index, having a marital partner, household income, gestational weight gain, and adverse pregnancy outcomes (low birth weight, small for gestational age, or hypertensive disorders of pregnancy).

**Table 6 nutrients-14-03505-t006:** Comparison of adverse outcomes between participants who received and those who did not receive iron supplementation during the first trimester.

	Iron or Multiple Micronutrients Supplementation during the First Trimester		*p*-Value
	Yes	No	
Participants who had anemia during the first trimester			
Low birth weight	1/34 (2.9)	7/48 (14.6)	0.13
Small for gestational age	5/34 (14.7)	10/48 (20.8)	0.48
Hypertensive disorders of pregnancy	1/34 (2.9)	2/48 (4.2)	1.00
K-EPDS ≥ 10 (postpartum)	9/25 (36.0)	9/41 (22.0)	0.21
HADS-A ≥ 8 (postpartum)	6/25 (24.0)	6/41 (14.6)	0.33
Participants who had normal hemoglobin levels during the first trimester			
Low birth weight	31/1200 (2.6)	81/1881 (4.3)	0.013
Small for gestational age	84/1200 (7.0)	167/1881 (8.9)	0.063
Hypertensive disorders of pregnancy	12/1200 (1.0)	22/1881 (1.2)	0.66
K-EPDS ≥ 10 (postpartum)	137/875 (15.7)	229/1373 (16.7)	0.52
HADS-A ≥ 8 (postpartum)	71/875 (8.1)	138/1373 (10.1)	0.12

HADS-A—anxiety score of Hospital Anxiety and Depression Scale; K-EPDS—Korean version of the Edinburgh Postnatal Depression Scale. The *p*-value was calculated using chi-squared test.

## Data Availability

All data are stored electronically in an anonymous format and are currently only available to KPOS researchers; however, data analysis collaborations may be possible through specific research proposals. Further information can be requested by e-mailing the corresponding author.

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
