# Peer review of "Maternal Anemia during the First Trimester and Its Association with Psychological Health"

_nutrients, 2022, doi:10.3390/nu14173505_

Round 1
Reviewer 1 Report
This is a very well conducted and written manuscript. Congratulations to the authors.
I have some minor comments that i have included in the manuscript and can be found in detail in the file attached.

Reviewer 2 Report
This manuscript presents a study that provides evidence of the impact of maternal anemia on pregnancy outcomes and maternal mental health. The study is well introduced, and the statistical analysis I consider to be adequate. I would like to congratulate the authors who prepared this manuscript.
However, there are some minor issues that the authors could address:
1. The methods section could be structured according to the STROBE guidelines (https://www.equator-network.org/reporting-guidelines/strobe/)
2. Present more clearly the inclusion/exclusion criteria.
3. Describe in more detail the confounders included in the multivariate logistic regression
Author Response
We appreciate your kind comments.
At first, editor pointed out that the length of the present version is a little shorter than what we expected for article paper. For this reason, we have added new content regarding “the effect of iron supplementation during the first trimester on pregnancy outcomes and psychological health at postpartum” to the Materials and Methods, Results, and Discussion section to increase the length and improve the quality of our manuscript. We believe that the newly added details will provide more informative insights for readers and improve the quality of our manuscript.
Hereafter, we answer the questions that you pointed out one by one.
Point 1: The methods section could be structured according to the STROBE guidelines (https://www.equator-network.org/reporting-guidelines/strobe/)
Response 1: As the reviewer’s suggested, we have separated the Materials and Methods section according to the STROBE guidelines.
Point 2: Present more clearly the inclusion/exclusion criteria.
Response 2: According to the review’s suggestion, we have added more information about the inclusion and exclusion criteria regarding the KPOS in the revised manuscript. Additionally, we have added the following information for the readers: “More details about the study design and participants of KPOS were described in our previous study [19].”
Point 3: Describe in more detail the confounders included in the multivariate logistic regression
Answer) As the reviewer’s suggested, we have added more details on the variables (confounders) used in multivariate logistic regression in the Materials and Methods section of the revised manuscript.